# Constrained DRL for Energy Efficiency Optimization in RSMA-Based Integrated Satellite Terrestrial Network

**DOI:** 10.3390/s23187859

**Published:** 2023-09-13

**Authors:** Qingmiao Zhang, Lidong Zhu, Yanyan Chen, Shan Jiang

**Affiliations:** 1National Key Laboratory of Science and Technology on Communications, University of Electronic Science and Technology of China, Chengdu 611731, China; qingmiao.zhang@outlook.com; 2School of Computer Science and Information Engineering, Xiamen Institute of Technology, Xiamen 361021, China; chenyanyan@xit.edu.cn; 3China Mobile (Jiangxi) Communications Group Co., Ltd., Yichun 336000, China; jiangshan@jx.chinamobile.com

**Keywords:** integrated satellite terrestrial network, rate splitting multiple access, energy efficiency, constrained deep reinforcement learning, soft actor-critic

## Abstract

To accommodate the requirements of extensive coverage and ubiquitous connectivity in 6G communications, satellite plays a more significant role in it. As users and devices explosively grow, new multiple access technologies are called for. Among the new candidates, rate splitting multiple access (RSMA) shows great potential. Since satellites are power-limited, we investigate the energy-efficient resource allocation in the integrated satellite terrestrial network (ISTN)-adopting RSMA scheme in this paper. However, this non-convex problem is challenging to solve using conventional model-based methods. Because this optimization task has a quality of service (QoS) requirement and continuous action/state space, we propose to use constrained soft actor-critic (SAC) to tackle it. This policy-gradient algorithm incorporates the Lagrangian relaxation technique to convert the original constrained problem into a penalized unconstrained one. The reward is maximized while the requirements are satisfied. Moreover, the learning process is time-consuming and unnecessary when little changes in the network. So, an on–off mechanism is introduced to avoid this situation. By calculating the difference between the current state and the last one, the system will decide to learn a new action or take the last one. The simulation results show that the proposed algorithm can outperform other benchmark algorithms in terms of energy efficiency while satisfying the QoS constraint. In addition, the time consumption is lowered because of the on–off design.

## 1. Introduction

As users and devices explosively grow in the next generation mobile communication systems and Internet of Things (IoT), it is more difficult to provide ubiquitous and reliable massive access with conventional terrestrial networks [1]. Since satellite communication has greatly developed over the years, it is considered a promising solution as a complement for terrestrial communications with its extensive coverage and continuous service [2]. However, it is still not enough to meet the demands with satellites alone; new multiple access (MA) technologies are also called for [3].

In long-term evolution (LTE), orthogonal frequency division multiple access (OFDMA) and singe carrier frequency division multiple access (SC-FDMA) are adopted to boost the capacity in downlink and uplink, respectively. By assigning users orthogonal sub-carriers or discrete Fourier transform (DFT) precoders, multiple access among them becomes possible. However, the orthogonality requirement limits their capacity. The massive machine type communication (mMTC) in 5G and 6G cannot be supported by them. Several non-orthogonal multiple access (NOMA) technologies are proposed to further improve the spectral efficiency. Among them, power domain NOMA (PD-NOMA) has been extensively studied over the years. Different transmission powers are allocated based on users’ channel conditions. By utilizing the successive interference cancellation (SIC) technique, multiple users could share the same frequency band and time slot, resulting in capacity boost. Nonetheless, it requires user ordering and significant channel discrepancy for better performance. The increase of users and devices also makes it more difficult and complex for user paring, ordering and power allocation, causing performance deterioration. Recently, a novel rate splitting multiple access (RSMA) has drawn a lot attention. It divides user’s messages into a public part and a private part, assigning them corresponding precoders. Users also share the same frequency and time resources. But it does not need user paring and ordering or great channel differences to perform well. It has higher capacity compared with OFDMA, SC-FDMA, and PD-NOMA with relatively low receiver complexity [4,5].

The idea of rate splitting was first presented in a single input single output (SISO) system [6]. Over the years, it has been extended for multiple input single output (MISO) and multiple input multiple output (MIMO) systems [7,8]. In the groundbreaking works of rate splitting (RS), the authors thoroughly study this method in terms of sum-rate theoretically; besides the obvious larger achievable rate region, the results prove that RS not only significantly decrease the complexity of precoder design and user scheduling but also exhibits robustness with imperfect channel state information (CSI) compared with conventional MA schemes [7,8]. For the classical weighted sum-rate (WSR) maximization in RSMA, usually this non-convex problem is converted to a convex weighted minimum mean square error (WMMSE) to solve [9]. But recently, Ref. [10] has adopted a new deep learning approach called deep unfolding to simplify and improve the WMMSE algorithm in an integrated satellite terrestrial network (ISTN), while [11] proposes to use the block-coordinate descent algorithm for an intelligent reflecting surface (IRS) aided system, both of which could achieve a better performance. Another hot topic of RSMA is the max-min fairness in a multi-group or multi-beam scenario. Reference [12] uses semi-definite relaxation and a convex-concave procedure to attain better fairness. Another reference [13] introduces LogSumExp approximation and a generalized power iteration framework for beamforming. Lately, the application of RSMA for simultaneous wireless information and power transfer (SWIPT) has gained more attention due to exponential energy consumption growth. In a multigroup multicast system, the authors of [14] uses the successive convex approximation (SCA) algorithm to minimize the transmission power.

Since the satellite is powered by solar panels, its power consumption cannot be neglected. Even though we could adopt RSMA to increase the capacity of ISTN, it is also crucial to complete the task with the lowest power possible. As for the energy efficiency optimization for RSMA, some of the literature proposes SCA-based algorithms for this problem [15,16]. However, these approaches are either too complex or only approximate to optimal solutions. Deep reinforcement learning, on the other hand, is a model-free method that can find the optimal solution with no need of prior knowledge. Additionally, the ability to learn is considered a fundamental requirement in the next-generation communication systems. Naturally, DRL is a hopeful candidate solution.

Using DRL for energy efficiency optimization in communications and other systems has been researched over the years. These studies fall into two categories: one with “discrete DRL” and one with “continuous DRL”. The former could only deal with discrete action/state space problems, e.g., deep Q-network (DQN) and state action reward state action (SARSA). However, most of the time, energy efficiency optimization is a continuous problem. Therefore, to apply “discrete DRL” to it, the action/state space should be discretized first. For example, Ref. [17] proposes to combine centralized DQN with multi-agent DQN in a 5G cognitive heterogeneous network, Ref. [18] utilizes a SARSA-based algorithm for ultra-dense networks, a multi-armed bandit RL is used for the energy-efficient trajectory design of unmanned aerial vehicles (UAVs). The second kind of RL could tackle continuous action/state space tasks, like deep deterministic policy gradient (DDPG) and proximal policy optimization (PPO). So there is no discretization requirement in these schemes. Ref. [19] applies DDPG to jointly optimize computing and caching; however, the constraints in it are simplified and not shown in the algorithm design. To allocate resources for 5G radio access network (RAN), asynchronous advantage actor-critic (A3C) is exploited. To satisfy the constraints, Ref. [20] uses many modules, which increases the complexity.

Besides DRL, other approaches—model-based or model-free—are also widely used for energy efficiency optimization in various systems. In [21], the authors adopt supervised machine learning to forecast traffic load and evaluate energy efficiency in Beyond 5G (B5G) networks. In wireless sensor networks (WSNs), Ref. [22] proposes a reliable clustering and routing algorithm considering different factors to save energy. The authors of [23] designed a cluster-based hierarchical routing protocol forming minimum spanning trees among communicating nodes, reducing energy wastage and prolonging the network lifetime in WSNs.

Energy efficiency optimization problems in communication networks are always limited by some conditions. Therefore, DRL, which could meet the constraints and tackle continuous tasks, is needed. Fortunately, constrained DRL (CDRL) has been proposed for this kind of problems. Generally, there are three main classes of CDRL:Primal-dual approach incorporates Lagrangian relaxation to transform the original constrained problem to an unconstrained dual problem [24,25];Policy optimization approach directly uses a more trackable function to replace original objective or constraint [26,27];Penalty function approach designs an unconstrained problem by adding penalty in the objective [28].

CDRL has been applied for network slicing recently. In [29], interior-point policy optimization (IPO) is adopted while [30,31] utilize the soft actor-critic (SAC) method. Another reference [32] gives an example of optimal power flow using constrained DDPG.

Nonetheless, there is no prior research on the energy efficiency optimization of RSMA adopting CDRL. So we propose to use SAC with a Lagrangian relaxation technique to improve the energy efficiency with the QoS constraint in ISTN. The contributions of our work are summarized as follows:We formulate the constrained Markov decision process (CMDP) framework of energy efficiency optimization, presenting a dual unconstrained problem adopting Lagrangian relaxation;We propose a constrained SAC algorithm whose objective function includes QoS constraint and entropy as a penalty;An on–off mechanism is introduced to avoid unwanted learning process when little changes in ISTN.

## 2. System Model

### 2.1. RSMA Architecture

To better understand how RSMA works, we use a *k*-user MIMO system to demonstrate the process, as shown in Figure 1. First, all users’ messages are divided into two parts: public and private parts, which would be $\left\{M_{1}^{\mathrm{pub}},M_{2}^{\mathrm{pub}},\cdots ,M_{k}^{\mathrm{pub}}\right\}$ and $\left\{M_{1}^{\mathrm{pri}},M_{2}^{\mathrm{pri}},\cdots ,M_{k}^{\mathrm{pri}}\right\}$ in the figure. Then, all the common messages are combined together to obtain one new message $M^{\mathrm{pub}}$ whereas *k* private messages remain untouched. By encoding, we can obtain k+1 streams $\left\{s_{\mathrm{pub}},s_{1},\cdots ,s_{k}\right\}$. Stream $s_{\mathrm{pub}}$ will be encoded again by a public precoder $w_{\mathrm{pub}}$ shared among all users. As for the private streams, each of them will be encoded by its own precoder $\left\{w_{1},w_{2},\cdots ,w_{k}\right\}$ accordingly. Finally, by superimposition, we will get the transmitted signal $w_{\mathrm{pub}}s_{\mathrm{pub}}+w_{1}s_{1}+\cdots +w_{k}s_{k}$.

At the receiver end, since the public precoder $w_{\mathrm{pub}}$ is known to all users, they can directly decode the signal to restore their public messages. Performing one-time successive interference cancellation (SIC) and decoding, one can restore its own private message by discarding all others’ private streams. After combination, the original message is obtained.

According to reference [9], RSMA is the bridge that connects spacial division multiple access (SDMA) and PD-NOMA. In SDMA, users are assigned with different beamforming precoders. Multi-user interference is discarded as noise. The performance of SDMA heavily depends on the orthogonality of users’ channels and the number of antenna.

In PD-NOMA, users are distinguished by different power. And in a multi-user system, the *k*-th user needs to perform (k−1) times SIC to restore its own message. That means multi-user interference needs to be fully decoded. PD-NOMA’s performance highly relies on the discrepancy of users’ channel gains and the perfect SIC operation.

Unlike these two extreme approaches, RSMA partially decodes the public stream and partially discards the private streams interference. This design allows it to achieve larger capacity. Moreover, the performance of RSMA depends on the precoder design and does not need channels to be orthogonal or significantly different. Therefore, RSMA is less limited than the other two. In addition, the one-time SIC in RSMA greatly reduces the complexity at receiver end. The advantages of RSMA could be summarized as follows.

It has a larger achievable rate region, which means it allows more connections;It has no requirements for users’ channel condition, so it could be applied to more scenarios;It does not need user pairing or ordering for SIC, which reduces the complexity and delay;It only needs one-time SIC operation. Complexity is reduced again and the error accumulation is avoided.

### 2.2. Network Model

The integrated satellite terrestrial network we investigate is depicted in Figure 2. In the downlink scenario, a satellite serves multiple users within its coverage. It could be a low earth orbit (LEO) satellite, a medium earth orbit (MEO) satellite or a geosynchronous (GEO) satellite. It is equipped with *L* antennas to provide extensive coverage. Each user is equipped with a single antenna. In this network, the RSMA scheme is adopted for multi-user transmission. Using different precoders, it can simultaneously serve multiple users and reuse the entire spectrum effectively.

To better describe the link between satellite and terrestrial users in real life, we employ the shadowed-Rician fading as our channel model [33]. Assume $h_{k}$ is the channel fading coefficient between satellite and *k*-th user. The probability density function (PDF) is listed below: (1)$$f_{{\left|h_{k}\right|}^{2}}\left(x\right)={\alpha_{k}e^{-\beta_{k}x}}_{1}F_{1}\left(m_{k};1;\delta_{k}x\right),$$
where $\alpha_{k}={\left(2b_{k}m_{k}/\left(2b_{k}m_{k}+\Omega_{k}\right)\right)}^{m_{k}}/2b_{k}$, $\beta_{k}=1/2b_{k}$, and $\delta_{k}=\Omega_{k}/2b_{k}\left(2b_{k}m_{k}+\Omega_{k}\right)$. $\Omega_{k}$ and $2b_{k}$ indicates the average power of line-of-sight (LoS) and multipath components, respectively. $m_{k}$ is the Nakagami-*m* parameter with range $\left[\left.0,\infty \right)\right.$. ${}_{1}F_{1}\left(a;b;x\right)$ is the confluent hypergeometric function given by
(2)$${}_{1}F_{1}\left(a;b;x\right)=\sum_{i=0}^{\infty }\frac{{\left(a\right)}_{i}}{{\left(b\right)}_{i}}\frac{x^{i}}{i!},$$
where ${\left(a\right)}_{i}=\Gamma \left(a+i\right)/\Gamma \left(a\right)$ is the Pochhammer symbol and $\Gamma \left(a\right)$ is the Gamma function [34].

According to [8], RSMA’s performance relies on the quality of CSI. In ISTN, the CSIs of satellites are continuously collected to monitor network condition and adjust precoders. However, perfect CSIs are unlikely to be obtained due to various reasons. A sensor could be introduced to assist CSI collection, but its placement will affect the results. Several papers [35,36,37] have studied this problem and proposed various algorithms. In this paper, for simplicity, we assume CSIs are acquired by terrestrial terminals.

The errors in CSI have various sources, such as Doppler shift [38] and satellite attitude dynamics [39,40]. Taking all these factors into account will cause rapid expansion of state space in DRL, which will lead to unstable learning and incapability to converge. Fortunately, the influence of these factors can be reflected in CSI. To achieve a tradeoff between complexity and accuracy, we add noise sampled from complex Gaussian distribution into the channel to simulate errors.

The satellite in this network divides the messages for all users and encodes them with RSMA precoders. After broadcasting, terrestrial users use a shared public precoder and their own private ones to restore the messages for them. To maximize the capacity with minimum power consumption, we will investigate the energy efficiency optimization in the next section.

## 3. Problem Formulation

As mentioned before, we define the vector of user streams as $s≜\left[s_{\mathrm{pub}},s_{1},\cdots ,s_{K}\right]\in C^{K+1}$ and the precoder matrix is $w≜\left[w_{\mathrm{pub}},w_{1},\cdots ,w_{K}\right]\in C^{L\times \left(K+1\right)}$. The final linearly precoded signal that satellite broadcasts to *K* users is
(3)$$x=ws=w_{\mathrm{pub}}s_{\mathrm{pub}}+\sum_{k=1}^{K}w_{k}s_{k},$$
where $w_{\mathrm{pub}},w_{k}\in C^{L}$ are used to execute simultaneous downlink transmissions. The received signal has the following form: (4)$$y=h^{H}x+n,$$
where $h=\left[h_{1},h_{2},\cdots ,h_{K}\right]$ is the downlink channel matrix, $n∼CN\left(0,I_{K}\right)$ is the additive white Gaussian noise (AWGN) vector, which is normalized for brevity. For better understanding, we take user *k* as an example, its received signal will be
(5)$$y_{k}=h_{k}^{H}w_{\mathrm{pub}}s_{\mathrm{pub}}+h_{k}^{H}w_{k}s_{k}+h_{k}^{H}\sum_{i\neq k}w_{i}s_{i}+n_{k},$$
where item $h_{k}^{H}\sum_{i\neq k}w_{i}s_{i}$ is the multi-user interference and $n_{k}$ is the noise.

Without loss of generality, we assume the power of users’ streams are normalized as well, which is $E\left[{\mathrm{ss}}^{H}\right]=I$. This will help us to obtain simpler and clearer mathematical expressions. Therefor, the SINR of the public part of the signal for user *k* is
(6)$$\gamma_{k}^{\mathrm{pub}}=\frac{{\left|h_{k}^{H}w_{\mathrm{pub}}\right|}^{2}}{\sum_{i=1}^{K}{\left|h_{i}^{H}w_{i}\right|}^{2}+1}.$$

Usually, we can now acquire the rate with Shannon’s formula. But in this scenario, we want to guarantee that all *K* users could successfully receive the signal. That means the rate of the public part of the signal should not exceed the user with lowest public part rate, which yields the following equation: (7)$$R_{k}^{\mathrm{pub}}=\min_{k}\left\{\log_{2}\left(1+\gamma_{k}^{\mathrm{pub}}\right)\right\}.$$

Similarly, the SINR of private part of the signal for user *k* will be
(8)$$\gamma_{k}^{\mathrm{pri}}=\frac{{\left|h_{k}^{H}w_{\mathrm{pub}}\right|}^{2}}{\sum_{i\neq k}^{K}{\left|h_{i}^{H}w_{i}\right|}^{2}+1}.$$

For the private part of the signal, user *k* will discard others’ private streams. So there is no rate requirement like the public part. The rate of private part of the signal for user *k* is
(9)$$R_{k}^{\mathrm{pri}}=\log_{2}\left(1+\gamma_{k}^{\mathrm{pri}}\right).$$

Now we can get the final rate of user *k*, which is
(10)$$R_{k}=R_{k}^{\mathrm{pub}}+R_{k}^{\mathrm{pri}}.$$

The total power consumption of the broadcast signal is $|w_{\mathrm{pub}}s_{\mathrm{pub}}{|}^{2}+\sum_{k}^{}{\left|w_{k}s_{k}\right|}^{2}$. Since we have normalized the power of user streams, we can obtain the power consumption as $\mathrm{tr}\left({\mathrm{ww}}^{H}\right)$, in which $\mathrm{tr}\left(\cdot \right)$ is the trace of a matrix. By the definition of energy efficiency, the problem we aim to solve is given as
(11)$$\max_{w}\;\;EE=\frac{\sum_{k=1}^{K}R_{k}}{\mathrm{tr}\left({\mathrm{ww}}^{H}\right)+P_{c}}.$$
(12)$$s.t.\;\mathrm{tr}\left({\mathrm{ww}}^{H}\right)\leq P_{t},$$
(13)$$R_{k}\geq R^{\mathrm{th}},$$
where $P_{c}$ is the circuit power consumed in the satellite, $P_{t}$ is the maximum transmission power of the satellite and $R^{\mathrm{th}}$ is the required minimum rate.

Equation (Equation 11) is the sum energy efficiency in the network. Constraint (Equation 12) is the power limit, indicating that the transmit power cannot exceed the maximum power. Constraint (Equation 13) is the QoS constraint to guarantee successful transmission. Since this problem is non-convex and NP-hard, conventional approaches use SCA-based methods to approximate the optimal solution. However, the complexities of these approaches are very high, and the optimal solution sometimes cannot be obtained. So we consider adopting a model-free DRL method for this problem.

To satisfy (Equation 12), we can simply use the projection operation to limit the precoders, which can be expressed as: (14)$$Proj\left(w\right)=\left\{\begin{array}{cc} w & \mathrm{if}\;\mathrm{tr}\left({\mathrm{ww}}^{H}\right)\leq P_{t},\\ w/∥w∥ & \mathrm{otherwise}. \end{array}\right.$$

Nonetheless, Constraint (Equation 13) is difficult to incorporate in a conventional DRL algorithm. Naturally, we want to adopt constrained DRL to tackle this problem, which utilizes different methods to satisfy the constraint. As the soft actor-critic algorithm has shown excellent performance and has been able to deal with continuous action/state space tasks, we combine it with Lagrangian relaxation technique to optimize the energy efficiency.

## 4. Constrained SAC Algorithm

### 4.1. Constrained Markov Decision Process Formulation

In a normal DRL framework, the problem should be first defined by a turple $\left(S,A,P,r\right)$ to solve, which consists of state space S, action space A, transition probability *P* and reward *r*. Similarly, a constrained DRL also needs this turple but with an extra cost function C. So, we give the following definitions.

State space is a set of representations of the environment observed by the agent. These observations have all the relevant information the agent needs to make a decision. In the ISTN, the SINRs, rates and energy efficiency can be calculated according to the channel fading coefficients, thus, we select them as the state. The current state space is defined as: (15)$$s^{\left(t\right)}=\left\{h_{1}^{\left(t\right)},h_{2}^{\left(t\right)},\cdots ,h_{K}^{\left(t\right)}\right\}.$$

Action space is a set of all valid actions of choices available to the agent. In this scenario, precoders are allocated to users to calculate energy efficiency with channel fading coefficients. So the current action space is given as
(16)$$a^{\left(t\right)}=\left\{w\right\},$$
where w is defined in Section 3.

Reward is a feedback from the environment based on the last action that agent took. It utilizes this feedback to improve its action policy. Usually, we can directly use energy efficiency as a reward; however, the QoS constraint cannot be incorporated in this case. Therefore, we introduce the Lagrangian relaxation method to convert the original problem into an unconstrained one.

According to references [41,42], the following constrained optimization
(17)$$\max_{x}\;\;f\left(x\right).$$
(18)$$s.t.\;g_{i}\left(x\right)\geq 0,$$
is equivalent to the problem
(19)$$\min_{\lambda >0}\;\max_{x}\;\hat{f}\left(x,\lambda ,\rho \right)=f\left(x\right)+\sum_{i}\lambda_{i}g_{i}\left(x\right),$$
where $\hat{f}\left(x,\lambda \right)$ is the Lagrangian relaxed form of original $f\left(x\right)$. The Lagrangian multiplier $\lambda_{i}$ adjusts the relative importance of constraint $g_{i}\left(x\right)$, which is called a cost function as well, against the objective function. Then, with the following equations at the *t*-th iteration, it can be gradually updated.
(20)$$\lambda_{i}^{\left(t+1\right)}=\lambda_{i}^{\left(t\right)}+\mu_{\lambda }g_{i}(x^{\left(t\right)}),$$
where $\mu_{\lambda }$ is the learning rate of the Lagrangian multiplier.

With the above information, the current instantaneous reward in Lagrangian relaxed form is defined as: (21)$$r\left(s^{\left(t\right)},a^{\left(t\right)}\right)=EE+\sum_{k=1}^{K}\lambda_{k}^{\left(t\right)}\left(R_{k}^{\left(t\right)}-R^{\mathrm{th}}\right),$$
in which the cost function is denoted as $c_{k}^{\left(t\right)}=R_{k}^{\left(t\right)}-R^{\mathrm{th}}$. The reward will decline when the QoS constraint is not satisfied.

### 4.2. The Proposed Constrained SAC Algorithm

Standard SAC is a policy gradient approach which uses the actor network to evaluate the policy and critic network to improve the policy. Additionally, the entropy of the policy is used as a penalty to improve its exploration, which yields fast convergence speed [43]. The maximum objective with entropy in it could be achieved when the optimum policy is found, which can be expressed as: (22)$$\pi^{*}=\underset{\pi }{\arg\max}\sum_{t=0}^{T}E_{s^{\left(t\right)},a^{\left(t\right)}}\left[r\left(s^{\left(t\right)},a^{\left(t\right)}\right)+\mu_{\pi }H\left(\pi \left(a^{\left(t\right)}|s^{\left(t\right)}\right)\right)\right],$$
where $\mu_{\pi }$ is the temperature parameter that regulates the relative importance of the entropy, which could increase the variance of the policy distribution so that more actions are available to be chosen. The Shannon entropy of the policy is defined as: (23)$$H\left(\pi \left(a^{\left(t\right)}|s^{\left(t\right)}\right)\right)=-\sum_{a^{\left(t\right)}}\pi \left(a^{\left(t\right)}|s^{\left(t\right)}\right)\log\pi \left(a^{\left(t\right)}|s^{\left(t\right)}\right).$$

The architecture of the proposed constrained SAC is illustrated in Figure 3. As shown in the picture, mainly three different components with five neural networks are used in this scheme, which are elaborated below.

To evaluate the soft state value, a deep neural network (DNN) with a target network is adopted, which are represented by parameters ψ and $\bar{\psi }$, respectively. The soft state value is used for soft Q-value approximation, both of which are utilized to evaluate received return in the future and are defined as: (24)$$V\left(s^{\left(t\right)}\right)=E_{a^{\left(t\right)}∼\pi }\left[Q\left(s^{\left(t\right)},a^{\left(t\right)}\right)-\log\pi \left(a^{\left(t\right)}|s^{\left(t\right)}\right)\right],$$
(25)$$Q\left(s^{\left(t\right)},a_{t}\right)=r\left(s^{\left(t\right)},a_{t}\right)+\rho E_{s^{\left(t+1\right)}∼P}\left[V\left(s^{\left(t+1\right)}\right)\right],$$
where ρ is the discount factor. With the objective and its gradient below, we can update the soft value network using the gradient descent method. This network is trained to minimize the square residual error with a target network to stabilize the training.
(26)$$J_{V}\left(\psi \right)=E_{s^{\left(t\right)}}\left[\frac{1}{2}{\left(V_{\psi }\left(s^{\left(t\right)}\right)-E_{a_{t}}\left[Q_{\theta }\left(s^{\left(t\right)},a^{\left(t\right)}\right)-\log\pi_{\phi }\left(a^{\left(t\right)}|s^{\left(t\right)}\right)\right]\right)}^{2}\right],$$
(27)$$\nabla J_{V}\left(\psi \right)=\nabla_{\psi }V_{\psi }\left(s^{\left(t\right)}\right)\left[V_{\psi }\left(s^{\left(t\right)}\right)-Q_{\theta }\left(s^{\left(t\right)},a^{\left(t\right)}\right)+\log\pi_{\phi }\left(a^{\left(t\right)}|s^{\left(t\right)}\right)\right].$$

The soft Q-value is estimated by double soft Q-value network. The two-Q-network design is meant to avoid overestimation. Parameters in this network could be updated by minimizing the soft Bellman residual. The objective and its gradient are expressed as: (28)$$J_{Q}\left(\theta \right)=E_{s^{\left(t\right)},a^{\left(t\right)}}\left[\frac{1}{2}{\left(Q_{\theta }\left(s^{\left(t\right)},a^{\left(t\right)}\right)-\left(r\left(s^{\left(t\right)},a^{\left(t\right)}\right)+\rho E_{s^{\left(t\right)}∼P}\left[V_{\bar{\psi }}\left(s^{\left(t+1\right)}\right)\right]\right)\right)}^{2}\right],$$
(29)$$\nabla J_{Q}\left(\theta \right)=\nabla_{\theta }Q_{\theta }\left(s^{\left(t\right)},a^{\left(t\right)}\right)\left[Q_{\theta }\left(s^{\left(t\right)},a^{\left(t\right)}\right)-r\left(s^{\left(t\right)},a^{\left(t\right)}\right)-\rho V_{\bar{\psi }}\left(s^{\left(t+1\right)}\right)\right].$$

Finally, the policy network provides the action for agent based on the optimal policy it finds. By minimizing the Kullback–Leibler (KL) divergence between the policy and soft Q-value, the parameters could be learned. The objective and its gradient are listed below.
(30)$$J_{\pi }\left(\phi \right)=E_{s^{\left(t\right)},a^{\left(t\right)}}\left[\log\pi_{\phi }\left(a^{\left(t\right)}|s^{\left(t\right)}\right)-Q_{\theta }\left(s^{\left(t\right)},a^{\left(t\right)}\right)\right],$$
(31)$$\nabla J_{\pi }\left(\phi \right)=\nabla_{\phi }\pi_{\phi }\left(a^{\left(t\right)}|s^{\left(t\right)}\right)+\left[\nabla_{a^{\left(t\right)}}\log\pi_{\phi }\left(a^{\left(t\right)}|s^{\left(t\right)}\right)-\nabla_{a^{\left(t\right)}}Q_{\theta }\left(s^{\left(t\right)},a^{\left(t\right)}\right)\right]\nabla_{\phi }f_{\phi }\left(\epsilon_{t};s^{\left(t\right)}\right),$$
where $a^{\left(t\right)}=f_{\phi }\left(\epsilon_{t};s^{\left(t\right)}\right)$ is the reparameterization, which could lower the variance estimation. $\epsilon_{t}$ is an input noise sampled from Gaussian distribution. The update process of λ could also be realized by neural networks; however, this will slow the learning process and destablize the network. So, we use the gradient ascend method for the update.

Moreover, since the learning process is time-consuming, if CSI changes little in the network, it is wasteful to learn a new action that almost equals to the last one. So we introduce an on–off mechanism to avoid unnecessary learning. When the difference between the current state and the last state is small enough, we will adopt the last action instead of learning a new one. With all the information above, we present our constrained SAC in Algorithm 1.
**Algorithm 1:** Constrained SAC algorithm.  1:initialize network parameters: ψ,θ,ϕ  2:initialize Lagrangian multiplier and its learning rate, temperature parameter and discount factor: $\lambda ,\mu_{\lambda },\mu_{\pi },\rho$  3:empty replay memory D, initialize last state $\bar{s}=0$ and initialize last action $\bar{a}=0$  4:copy parameter to target network: $\bar{\psi }\leftarrow \psi$  5:**for** each episode (t=1,2,⋯,T) **do**  6: observe environment $s^{\left(t\right)}$  7: compute the difference between $s^{\left(t\right)}$ and $\bar{s}$: $\Delta =\left|s^{\left(t\right)}-\bar{s}\right|$.  8: **if** Δ is greater than the threshold **then**  9:  sample action $a^{\left(t\right)}$ from policy network: $a^{\left(t\right)}∼\pi_{\phi }\left(\cdot |s^{\left(t\right)}\right)$10:  set $\bar{s}=s^{\left(t\right)}$ and $\bar{a}=a^{\left(t\right)}$11: **else**12:  take action $a^{\left(t\right)}=\bar{a}$13: **end if**14: calculate reward according to Equation (Equation 21)15: update replay memory: $D\leftarrow D\cup \left(s^{\left(t\right)},a^{\left(t\right)},r,s^{\left(t+1\right)},c_{k}^{\left(t\right)}\right)$16: **if** it is time to update **then**17:  sample a mini batch from D.18:  update ψ via stochastic gradient ascent on (Equation 27)19:  update θ via stochastic gradient ascent on (Equation 29)20:  update ϕ via stochastic gradient ascent on (Equation 31)21:  update λ according to (Equation 20)22:  update $\bar{\psi }$ by $\bar{\psi }\leftarrow \tau \psi +\left(1-\tau \right)\bar{\psi }$23: **end if**24:**end for**

## 5. Simulation Results

In order to verify the performance of the proposed constrained SAC algorithm, we test it in various scenarios with different algorithms, including a DDPG approach incorporating penalty as constraint in its reward function, unconstrained SAC, and conventional SCA-based algorithm.

### 5.1. Simulation Parameters

In ISTN, multiple users are served by an LEO satellite within its coverage. The free space loss coefficient is given as $\eta =\left(1/4f\pi d\right)$, in which *f* and *d* are the frequency and distance between satellite and user.

The neural networks (NNs) in the proposed algorithm all consist of three fully connected hidden layers with ReLU as the activation function and Adam optimizer. The first two hidden layers of NN in the soft value network has 64 neurons while the last layer has 1 neuron. The hidden layers of NNs in the policy network and the soft Q network all have 64 neurons. The learning rate of the Adam optimizer is $10^{-3}$ while the update rate of target networks is 0.005. Their loss functions are given in Equations (Equation 21), (Equation 22) and (Equation 27), respectively. Replay memory size and batch size are 200 and 32, respectively, and the episode is 5000 with 200 timestamps. Since all users are equally “important” in the ISTN, their QoS constraints contribute to the reward equally as well. So, we initialize the Lagrangian multipliers to 1. The initial value of the learning rate of the Lagrangian multiplier is usually small. So we set it to be $10^{-3}$. The tuning of them could be performed by introducing additional NNs with different optimizers; however, we fix their values to avoid extra complexity. The parameters are listed in Table 1.

### 5.2. Results and Discussion

We evaluate the average reward of constrained SAC with different users, which is depicted in Figure 4. Clearly, it converges in all four scenarios around 200 to 300 episodes, and its average reward increases as users grow in the ISTN. Since we initialize the Lagrangian multiplier as 1 and the rate constraint as 4. The start points of four lines are −8, −12, −16 and −20, respectively. Two things need to be noted. The first is that the fluctuation becomes greater when users grow. The reason for this is that when more users served in the network, the action space becomes larger, making it more difficult to design the precoders for each user. Another thing is that the gap between two lines shrinks as users increase. Given the transmission power of the satellite, the achievable rate of each user will slightly decrease when more users join the network, resulting in slowing the growth speed of the average reward.

The energy efficiency performance of constrained SAC with different users is shown in Figure 5. We can see that it shares some similarities with Figure 4, such as the greater fluctuation and smaller gap with more users. The reasons are the same—more users enlarge the action space. Since energy efficiency is the reward without constraint, the start points of these four lines are all 0.

Next, we verify whether the rate constraint is satisfied with the proposed algorithm. In Figure 6, we can see that with different users, constrained SAC meets all the demands. For clarity, we add four straight lines to indicate the rate constraints in four scenarios. The capacity generated by the proposed algorithm all exceed the constraints. With these three results, the constrained SAC can maximize the energy efficiency under the rate constraint. It provides an alternative solution for constrained optimization problem.

To evaluate the performance of constrained SAC (denoted as CSAC in the figures) with other benchmark algorithms, we choose the original SAC with no constraint and DDPG incorporating constraint into its reward function (denoted as DDPG in the figures) [44]. The existing model-based SCA optimization (denoted as SCA in the figures) is compared with these model-free DRL methods as well [15,16].
(32)$$r_{\mathrm{DDPG}}^{\left(t\right)}=EE\left(1-\eta^{\left(t\right)}\right),$$
where $\eta \left(t\right)$ is the penalty defined as
(33)$$\eta^{\left(t\right)}=\frac{1}{K}\sum_{i=1}^{K}I\left(R_{k}^{\left(t\right)}-R^{\mathrm{th}}\right),$$
in which $I\left(\cdot \right)$ is a indicator function given by
(34)$$I\left(x\right)=\left\{\begin{array}{cc} 1 & x<0\\ 0 & x\geq 0 \end{array}\right..$$

Whenever the rate is lower than the threshold, the reward of DDPG will be penalized. Therefore, to achieve the maximum reward, the rate will meet the requirement.

In the three users scenario, since the original SAC is unconstrained, this means its reward is the energy efficiency. Therefore, it looks like a straight line compared with the other two algorithms that have a larger reward in Figure 7. Constrained DDPG, however, uses the constraint as penalty in its reward function, showing a similar look to constrained SAC. But it is obvious that its convergence speed is slower. The reason for this is that constrained SAC has entropy in the reward function and noise in its action for more exploration, which will eventually improve the convergence speed. Therefore, CSAC has a better exploration performance than DDPG, which only adds noise in the action. And the reward DDPG achieves is a bit lower than that of our algorithm.

The energy efficiency performance comparison is depicted in Figure 8. Original SAC has the lowest performance because its rate requirement cannot be satisfied. DDPG once again presents relatively slow convergence speed. Its energy efficiency is higher than original SAC but lower than constrained SAC. SCA converges after five iterations. Its energy efficiency is even lower than DDPG, demonstrating that DRL could achieve better performance than the model-based optimization approach without any prior knowledge. However, unlike the “black-box” characteristic of DRL, SCA has more explicit expression. CSAC outperforms the other three with the highest performance, which validates its effectiveness.

We also compare the capacities of four algorithms to check if the constraint satisfied. In Figure 9, the straight line represents the capacity requirement in the three users scenario, which is 12 bps/Hz. CSAC, DDPG, and SCA meet the capacity requirement with their different approaches. Original SAC, on the other hand, fails to satisfy the constraint, even though its fluctuation goes beyond the straight line in rare cases. CSAC achieves the highest capacity among them, which results in the highest energy efficiency in Figure 8.

We investigate the effect of the on–off mechanism. In Figure 10, we vary thresholds to observe the energy efficiency change of the three algorithms. As the threshold increases, it is more likely that the last action will be taken instead of finding a new one. Consequently, the energy efficiency will decrease as the action does not perfectly apply to the new state. According to [9], when the strength difference of two channels is 0.3, additional 5 dB loss will be introduced. Although less learning process with a bigger threshold will cost less time, the tradeoff between time cost and performance should be considered carefully. Normally, the threshold needs to be less than 0.3, or it will deteriorate the capacity and energy efficiency.

Figure 11 shows the time cost performance of three algorithms. All of their time consumption is lowered when adopting an on–off mechanism because some of the learning process is substituted by the actions from the last states. Among them, the original SAC has the lowest time consumption because it does not need to compute the Lagrangian multiplier or penalty. DDPG has a simpler architecture and constraint in its reward function than CSAC, thus the lower time cost.

Finally, the energy efficiency performance of CSAC and DDPG with different constraint requirements is illustrated in Figure 12. As the rate constraint grows, it is more difficult for CSAC and DDPG to find the suitable precoders, so their energy efficiency declines. However, the penalty design in DDPG is more sensitive to requirement change, and it dropps faster than CSAC.

## 6. Conclusions

We investigate the energy efficiency optimization of rate splitting multiple access in an integrated satellite terrestrial network, where multiple users are served by a satellite with a QoS constraint. This problem is non-convex and NP-hard; therefore, we adopt a soft actor-critic deep reinforcement learning approach as it can deal with continuous state/action space tasks. To satisfy the QoS constraint, the Lagrangian relaxation technique is incorporated to convert the original problem into a constrained one. Also, an on–off mechanism is introduced to avoid frequent unnecessary learning processes if the state changes little. The simulation results validate that the proposed constrained SAC algorithm achieves a better performance while meeting the QoS requirement comparing with other algorithms in terms of convergence speed and energy efficiency. Time cost is reduced as well when the on–off mechanism is adopted. Future work will focus on CSAC improvement, such as exploration and sample efficiency, and on RSMA in a more complicated system like integrated space air terrestrial networks and satellite-based sensor networks.

## Figures and Tables

**Figure 1 sensors-23-07859-f001:**
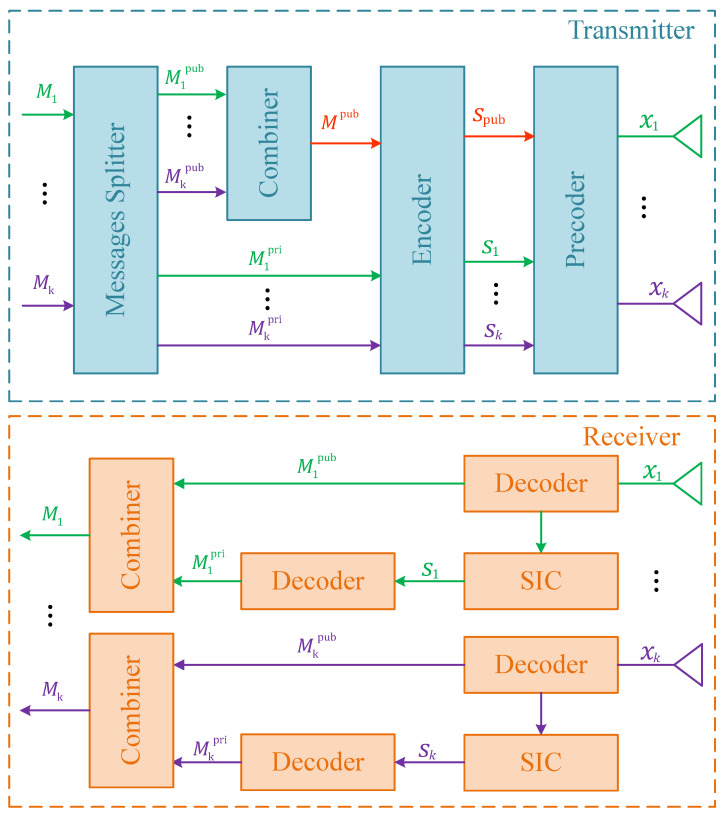
RSMA architecture.

**Figure 2 sensors-23-07859-f002:**
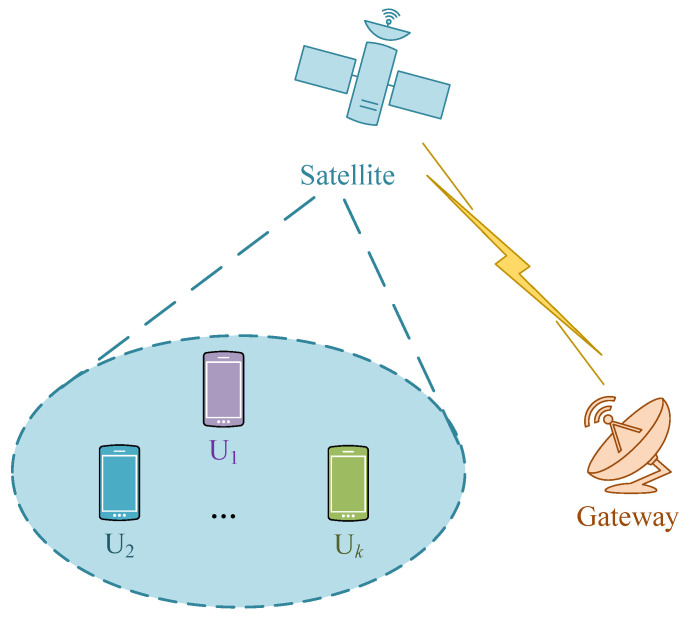
Network model.

**Figure 3 sensors-23-07859-f003:**
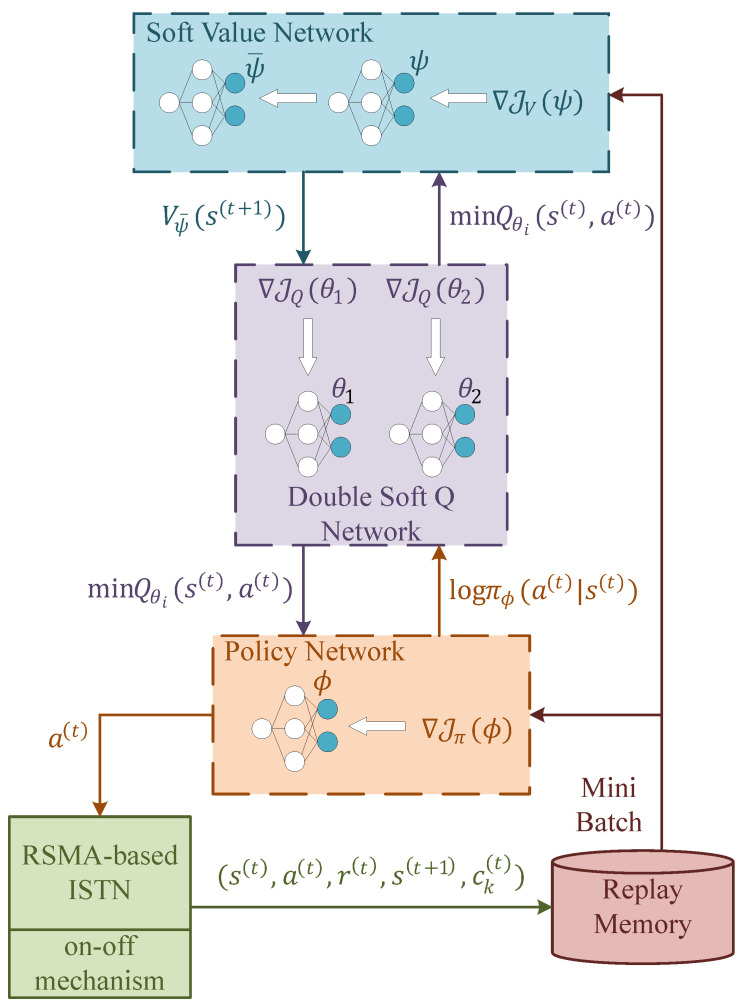
Architecture of constrained SAC.

**Figure 4 sensors-23-07859-f004:**
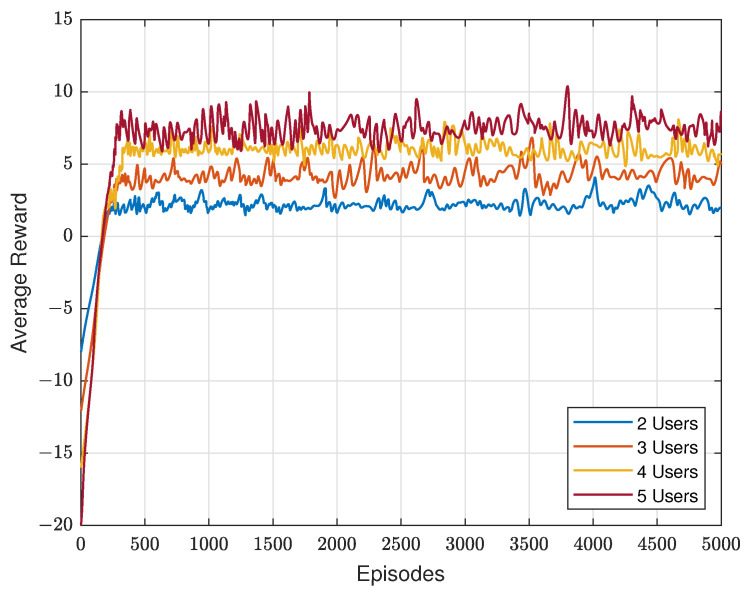
Average reward of constrained SAC with different users.

**Figure 5 sensors-23-07859-f005:**
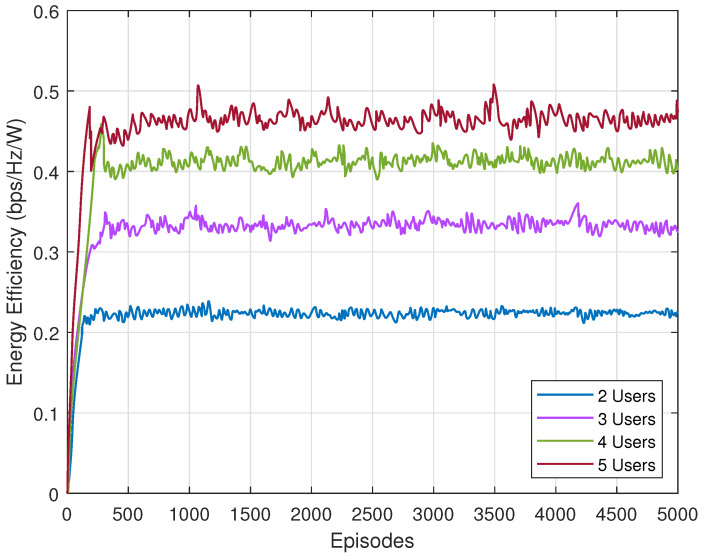
Energy efficiency of constrained SAC with different users.

**Figure 6 sensors-23-07859-f006:**
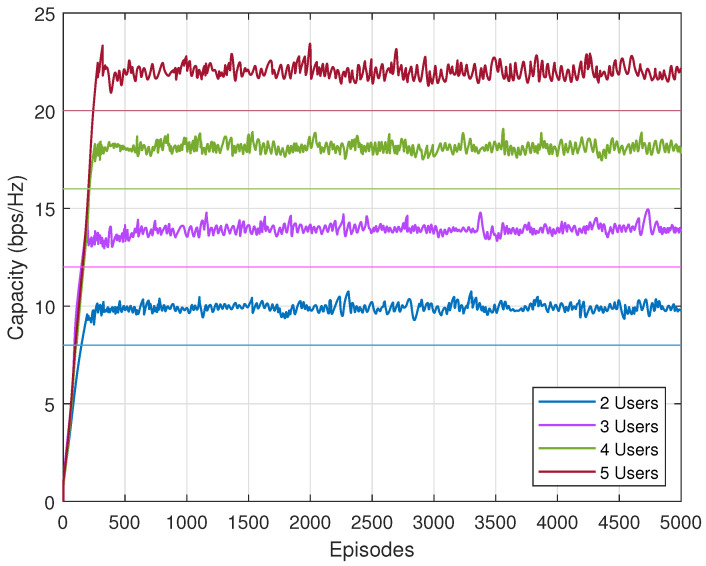
Capacity of constrained SAC with different users.

**Figure 7 sensors-23-07859-f007:**
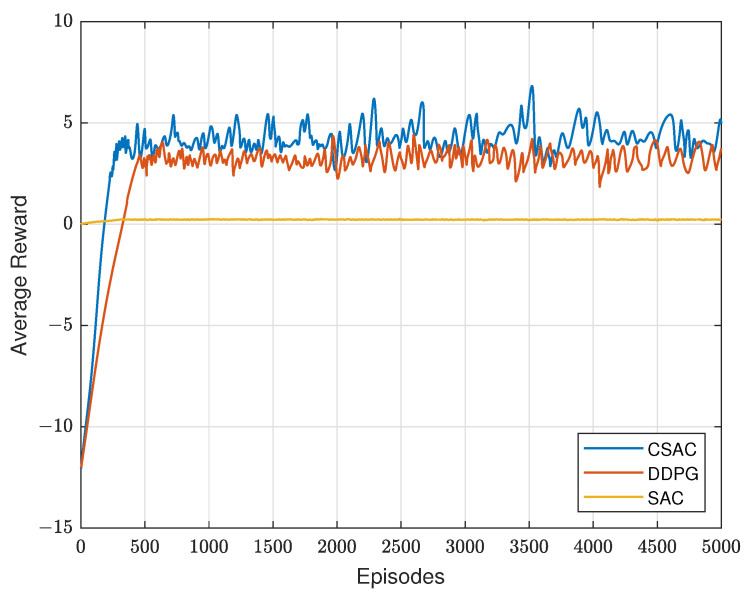
Average reward with different algorithms.

**Figure 8 sensors-23-07859-f008:**
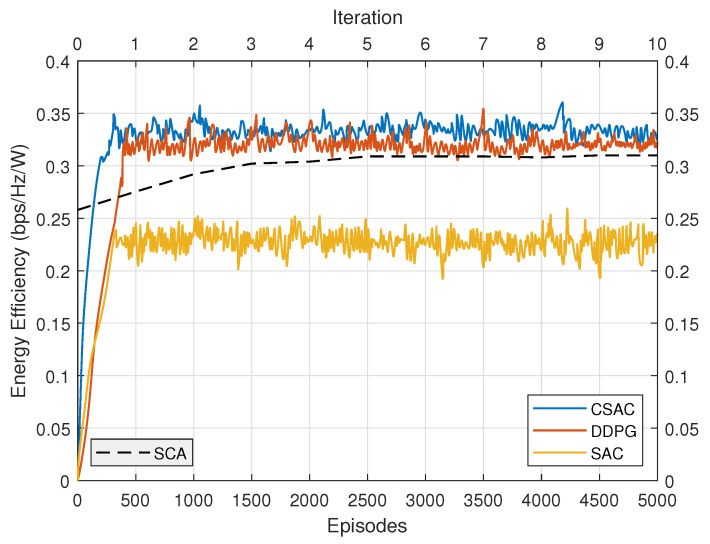
Energy efficiency with different algorithms.

**Figure 9 sensors-23-07859-f009:**
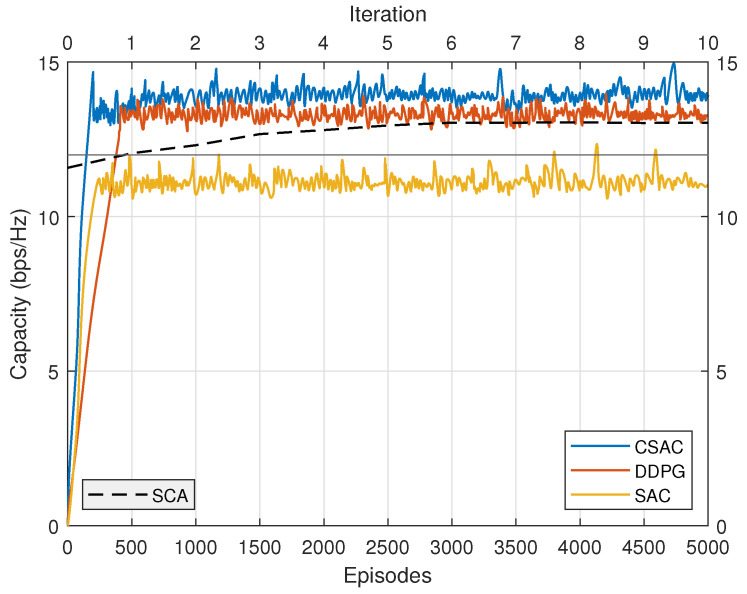
Capacity with different algorithms.

**Figure 10 sensors-23-07859-f010:**
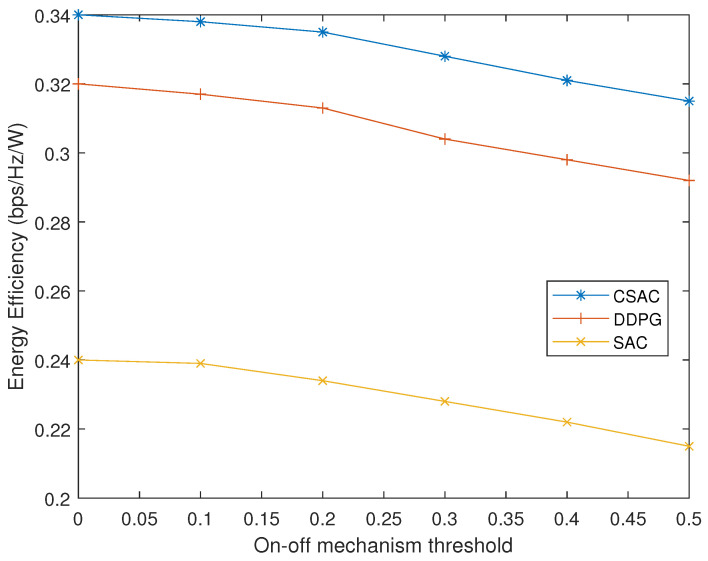
Energy efficiency with different on–off mechanism thresholds.

**Figure 11 sensors-23-07859-f011:**
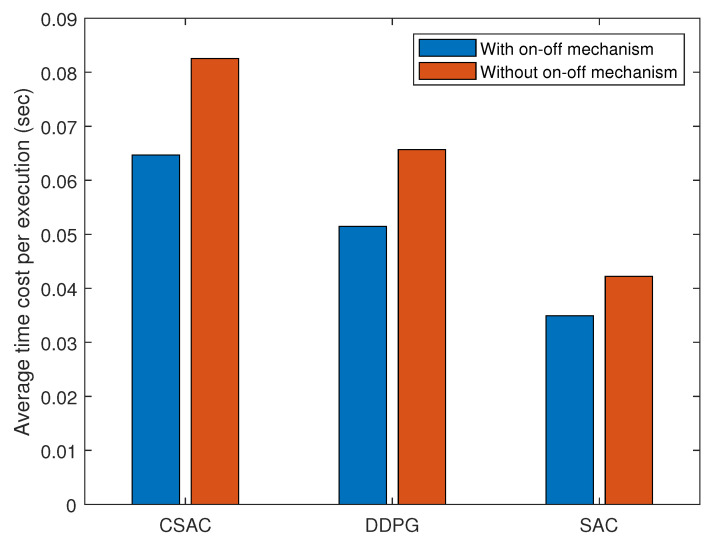
Average time cost per execution with different on–off mechanism thresholds.

**Figure 12 sensors-23-07859-f012:**
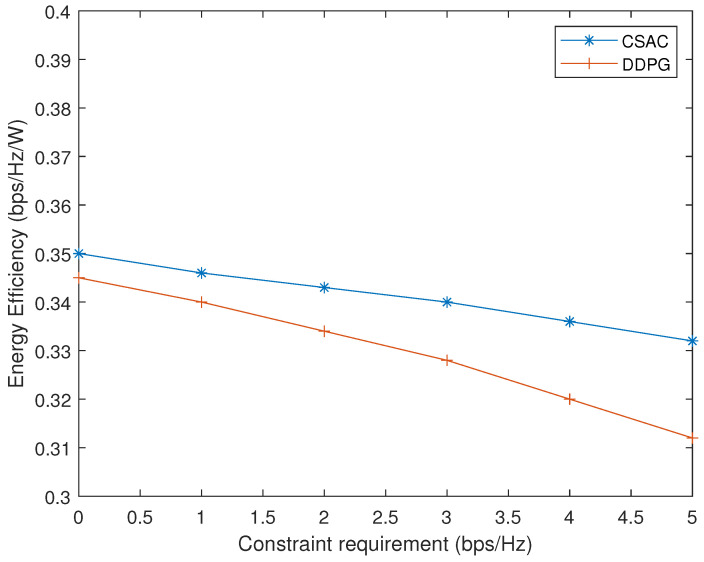
Energy efficiency with different constraint requirements.

**Table 1 sensors-23-07859-t001:** Simulation parameters.

Parameter	Value	Note
*b*	0.158	Average power of multi-path component
Ω	1.29	Average power of LoS component
*m*	19.4	Nakagami-*m* parameter
*f*	12 GHz	Satellite frequency
*d*	1000 km	Distance between satellite and users
*K*	2, 3, 4, 5	Number of users
$P_{t}$	40 W	Transmission power of satellite
$P_{c}$	5 W	Circuit power
Δ	0.1	Difference between last state and current state
$R^{\mathrm{th}}$	4 bps/Hz	Rate constraint
ρ	0.98	Discount factor
λ	1	Lagrangian multiplier
$\mu_{\lambda }$	$10^{-3}$	Learning rate of λ
$\mu_{\pi }$	0.2	Temperature parameter for entropy

## Data Availability

Data sharing not applicable.

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
