# Peer review of "Constrained DRL for Energy Efficiency Optimization in RSMA-Based Integrated Satellite Terrestrial Network"

_sensors, 2023, doi:10.3390/s23187859_

Round 1

Reviewer 1 Report

This study proposed to use constrained deep reinforcement learning in this scenario, which incorporates Lagrangian relaxation technique into policy-gradient soft actor-critic (SAC) algorithm. Overall, the manuscript is interesting and organized well with novelties, which can be accepted after minor revision.

1.       The language and grammar should be carefully checked. The articles (a, an, the) should be applied rightly.

2.       How to set the key parameters in the proposed method.

3.       How to place the measurement points? The recent advances on sensor placment which are directly related to the proposed study should be referred including “Sensor placement algorithm for structural health ……”, A novel load-dependent sensor placement method ……”, and “A novel two-step strategy of non-probabilistic multi-objective ……”.

4.       How many users can there be at most? How does the number of different users affect the efficiency of the proposed method?

5.       The source of the error should be introduced. Recent research progress on this topic should be referred including Reliability-constrained optimal attitude-vibration ……” and Uncertain optimal attitude control for space ……”.

6.       Future research plans can prospect in Conclusions.

Well

Author Response

Thank you very much for taking the time to review this manuscript. Please find the detailed responses below and the corresponding revisions in the re-submitted files.

Comments 1: The language and grammar should be carefully checked. The articles (a, an, the) should be applied rightly.

Response 1: We have carefully checked the paper and corrected the errors.

Comment 2: How to set the key parameters in the proposed method.

Response 2: The key parameters setting in the proposed method refers to several literatures [10.1109/TAI.1998.744838, 10.1109/LWC.2022.3172336, 10.3390/electronics11203385, 10.48550/arXiv.1812.05905]. Besides the ones in Table 1, we also add some additional information such as the architecture of neural networks that we didn’t mention before in the paper.

Comment 3: How to place the measurement points? The recent advances on sensor placement which are directly related to the proposed study should be referred including “Sensor placement algorithm for structural health ……”, “A novel load-dependent sensor placement method ……”, and “A novel two-step strategy of non-probabilistic multi-objective ……”.

Response 3: Thank you for pointing this out. In this paper, we want to focus on the multiple access used for transmission between satellite and terrestrial users. However, RSMA’s performance relies on the quality of channel state information (CSI). Sensors could be used to assist the collection of CSI. We will look into satellite-based sensor network adopting RSMA in the future. We have added this and the papers listed above in Section 2.2.

Comment 4: How many users can there be at most? How does the number of different users affect the efficiency of the proposed method?

Response 4: Due to the hardware limitation, at most 7 users scenario was simulated. As users grow in the network, the learning process became more and more time-consuming and unstable, even causing PC breakdown. Because the growth of users leads to rapid expansion of action and state space, it would require more resource for calculation. If number of users is smaller than 7, the efficiency of the proposed algorithm will slightly decline because of the enlarged action space.

Comment 5: The source of the error should be introduced. Recent research progress on this topic should be referred including “Reliability-constrained optimal attitude-vibration ……” and “Uncertain optimal attitude control for space ……”.

Response 5: Thank you for this suggestion. The error that could deteriorate RSMA’s performance mainly comes from the imperfect CSI. Other factors that might affect it could also be reflected in CSI. So, we add noise in the channel realization to simulate real life environment. This way we could achieve the tradeoff between complexity and accuracy, because too many influencing factors will also lead to rapid expansion of state space, making simulations difficult to be carried out. However, it is more accurate to take as many as possible influencing factors into account. So we will consider it in the future work to improve our research. The discussion and paper listed above have been added in Section 2.2.

Comment 6: Future research plans can prospect in Conclusions.

Response 6: We have added future research plans in conclusions, including the satellite-based sensor networks and improvement of CSAC.

Reviewer 2 Report

1. Reframe the first line of the abstract. Change it from a complex to a simpler one.

2. How RSMA differs from OFDMA and SCFDMA? This needs to be addressed in the introduction.

3. Compare your results with the existing literature to corroborate your works.

4. The abstract is very brief. Do expand it.

Minor editing is required. some sentences are complex

Author Response

Thank you very much for taking the time to review this manuscript. Please find the detailed responses below and the corresponding revisions in the re-submitted files.

Comments 1: Reframe the first line of the abstract. Change it from a complex to a simpler one.

Response 1: We have rewritten the entire abstract.

Comment 2: How RSMA differs from OFDMA and SCFDMA? This needs to be addressed in the introduction.

Response 2: To support multiple users, OFDMA assigns them orthogonal multiple sub-carriers while SCFDMA uses discrete Fourier Transform (DFT) precoders. Both of them depend on orthogonality to improve capacity. RSMA, on the other hand, designs the precoders aiming to reduce the multi-user interference to noise level. It does not need the precoders to be orthogonal, therefore, it can greatly boost the capacity in the network.

The comparison of RSMA with OFDMA and SCFDMA has been added in the introduction.

Comment 3: Compare your results with the existing literature to corroborate your works.

Response 3: We have added a successive convex approximation-based algorithm used for energy efficiency optimization in [15, 16] to compare with our method. The results are shown in Figure 8 and 9.

Comment 4: The abstract is very brief. Do expand it.

Response 4: The entire abstract has been rewritten.

Reviewer 3 Report

Technical Remarks:

  • Provide more details on the neural network architecture and hyperparameters used in the CDRL algorithm. What activation functions, a number of layers/units, loss functions, etc.
  • Explain how the Lagrangian multiplier and its learning rate were initialized and tuned.
  • Analyze the results more deeply, e.g., discuss why CSAC converges faster than DDPG, and how the on-off threshold affects performance tradeoffs.
  • Expand the related work section to differentiate the contributions of this paper from prior art.
  • Add these papers : https://doi.org/10.3390/s22228950 and https://doi.org/10.3390/s22228614 

Questions for Authors:

  1. How was the splitting of messages into public and private parts done? Was this optimally designed or just a preset arbitrary split?
  2. What prevents the CDRL algorithm from converging to a bad local optimum solution? How can exploration be improved?
  3. Could you explain more about the on-off mechanism? How exactly is the threshold adapted over time?
  4. What are some ways the CSAC algorithm could be extended or improved in future work?
